# Hedgehog Signaling in Myeloid Malignancies

**DOI:** 10.3390/cancers13194888

**Published:** 2021-09-29

**Authors:** Ajay Abraham, William Matsui

**Affiliations:** Department of Oncology, Dell Medical School, University of Texas at Austin, Austin, TX 78712, USA; ajay.abraham@austin.utexas.edu

**Keywords:** hedgehog, smoothened, leukemia, myeloid disorders, hematopoiesis

## Abstract

**Simple Summary:**

The Hedgehog signaling pathway is aberrantly activated in many myeloid malignancies, and pathway inhibition is clinically beneficial in specific patients with acute myeloid leukemia. However, even with the approval of these agents, the role of Hedgehog signaling in other myeloid disorders is less clear. In this review, we summarize the laboratory studies that have examined Hedgehog signaling in normal and malignant hematopoiesis as well as the clinical studies that have been carried out in several myeloid leukemias. Finally, we explore potential strategies to further expand the use of pathway inhibitors as therapies for these diseases.

**Abstract:**

Myeloid malignancies arise from normal hematopoiesis and include several individual disorders with a wide range of clinical manifestations, treatment options, and clinical outcomes. The Hedgehog (HH) signaling pathway is aberrantly activated in many of these diseases, and glasdegib, a Smoothened (SMO) antagonist and HH pathway inhibitor, has recently been approved for the treatment of acute myeloid leukemia (AML). The efficacy of SMO inhibitors in AML suggests that they may be broadly active, but clinical studies in other myeloid malignancies have been largely inconclusive. We will discuss the biological role of the HH pathway in normal hematopoiesis and myeloid malignancies and review clinical studies targeting HH signaling in these diseases. In addition, we will examine SMO-independent pathway activation and highlight potential strategies that may expand the clinical utility of HH pathway antagonists.

## 1. Introduction

The Hedgehog (HH) signaling pathway is highly conserved and required for normal embryonic development in both invertebrates and vertebrates [1,2]. Along with other developmental signaling pathways, including Notch and Wnt, it specifies both early patterning and polarity events and the subsequent formation of specific organs by spatially and temporally regulating cell proliferation and differentiation. Aberrant HH signaling has been identified in a wide range of malignancies, and translational efforts have led to the development of several pathway antagonists. One approved clinical indication for HH pathway inhibitors is the treatment of acute myeloid leukemia (AML), and in this review we will discuss the role of HH signaling in normal hematopoiesis and malignancies arising from hematopoietic stem cells (HSCs) as well as future strategies to expand the clinical utility of pathway inhibition.

## 2. Hedgehog Signaling and Hematopoiesis

The activation of the HH signaling pathway in mammals is initiated by binding one of the three HH ligands found in mammals, Sonic (SHH), Indian (IHH), or Desert (DHH), to the 12-pass transmembrane receptor Patched (PTCH). All three of these HH ligands are post-translationally modified by the covalent attachment of cholesterol and palmitate moieties that are required for proper secretion, extracellular distribution along morphogenic gradients, and maximal pathway activation [3]. In the unbound state, PTCH represses the activity of the 7-pass transmembrane signal transduction protein Smoothened (SMO) that shares homology with G protein-coupled receptors. The de-repression of SMO ultimately leads to the expression and/or post-translational processing of the three GLI zinc-finger transcription factors that regulate the expression of HH target genes. GLI1 is a transcriptional activator, GLI3 primarily serves as a transcriptional repressor, and GLI2 can either activate or repress gene expression depending on post-transcriptional splicing events and post-translational modifications [4,5]. The balance between the activating and repressive forms of the three GLI transcription factors dictates the overall expression of HH target genes, including *PTCH1* and *GLI1*, and the functional impact of pathway activity [4,6].

Hematopoiesis provides life-long blood cell production, and the HH signaling pathway has been found to regulate the development of the hematopoietic system during embryogenesis as well as definitive post-natal hematopoiesis [7]. Within primitive yolk sac hematopoiesis, IHH ligand expression leads to HH pathway activation within epiblasts to specify the patterning of the hematopoietic mesoderm and eventually the initial stages of hematopoiesis and angiogenesis. In a similar fashion, IHH or SMO deficiency impairs the formation of blood islands and organized vasculature by embryonic stem cells that can replicate the development of primitive hematopoiesis. In definitive hematopoiesis giving rise to all blood cells found in adults, the role of HH signaling has been difficult to understand because different model systems have provided conflicting results [8]. Initial studies demonstrated that primitive CD34^+^CD38^neg^Lin^neg^ human cord blood HSCs express *PTCH1*, *SMO*, and *GLI* transcription factors, and HH pathway activation with the exogenous Shh ligand induced proliferation in a BMP4-dependent manner [9]. Subsequent studies using mice with somatic or conditional loss of *Ptch1* expression to increase baseline HH pathway activity demonstrated an expansion of HSCs and improved hematopoietic recovery after bone marrow injury with 5-fluorouracil [10,11]. This sustained proliferation eventually led to HSC exhaustion, but treatment with cyclopamine, a small molecule inhibitor of SMO, restored long-term HSC function. In line with these results, the loss of *Smo* expression and HH signaling was found to profoundly impact HSC function during initial and secondary transplantation [12].

In contrast, multiple other studies demonstrated that the loss of *Smo* either in fetal liver HSCs or adult bone marrow HSCs or exposure to cyclopamine had no impact on the ability of HSCs to home to the bone marrow or provide long-term engraftment during serial transplantation. These studies concluded that Smo is completely dispensable for normal adult HSC function, suggesting that SMO inhibition will have little hematologic toxicity [11,13,14] Discrepancies between these findings may be due to the nature of the gene promoter (i.e., *Vav* or *Mx1*) used to conditionally express Cre recombinase and knockout *Smo* expression in hematopoietic cells. Given that GLI1 is the major effector of HH pathway activation, the loss of GLI1 in hematopoiesis has also been studied [15]. In somatic *Gli1* knockout mice, HSC proliferation was decreased within both HSCs and lineage-committed progenitors associated with lower expression of the Gli1 target gene Cyclin D1, and increased HSC quiescence improved both long- and short-term HSC engraftment. Therefore, the precise role of HH signaling in adult hematopoiesis remains unclear.

## 3. Hedgehog Signaling in Chronic Myeloid Leukemia

The HH pathway was initially implicated in cancer development by studies in Gorlin syndrome that demonstrated a high frequently of loss of function mutations in *PTCH1* that result in ligand-independent pathway activation [16]. Myeloid malignancies represent a heterogeneous group of disorders arising from normal HSCs and hematopoietic progenitors, and the HH pathway has been implicated in several of these diseases. Chronic myeloid leukemia (CML) is characterized by the Philadelphia chromosome that encodes the fusion BCR-ABL tyrosine kinase. The role of the HH pathway in CML was initially suggested by high *Ptch1* expression in CD34^+^ cells from blast crisis CML (CML-BC) clinical specimens [17]. Subsequent functional studies used CML mouse models in which bone marrow or fetal liver cells transduced ex vivo with the BCR-ABL gene give rise to leukemia following transplantation into naïve recipients [11,12]. Compared to normal hematopoietic cells, the transduction of BCR–ABL into Smo-deficient cells led to a significant decrease in disease burden and improved survival rates. Leukemic stem cells (LSCs) are enriched for clonogenic growth potential and self-renewal similar to normal HSCs and are thought to be responsible for resistance to BCR–ABL tyrosine kinase inhibitors (TKIs) and disease relapse. In CML, the expression of HH target genes, including *GLI1* and *PTCH1*, is low in leukemia cells isolated from patients in CML-CP, but increases during progression to CML-AP and CML-BC [11,18,19]. Multiple studies have demonstrated that HH signaling supports LSCs and cyclopamine, and other SMO antagonists can limit the functional properties of CML LSCs and resistance to TKIs [20,21,22,23]. Therefore, pharmacologic inhibition of the HH signaling pathway may reduce CML relapse.

The identification of aberrant HH signaling pathway activity in cancer has led to the development of pathway inhibitors as potential therapeutic agents. Initial evidence that the HH signaling pathway could be pharmacologically targeted was provided by the discovery that cyclopamine, a steroidal alkaloid found in *Veratum californicum*, was a naturally occurring inhibitor of SMO. Cyclopamine was originally identified as the causal agent of congenital holoprosencephaly in sheep ingesting *V. californicum*, and the genetic loss of SHH in mice resulted in similar morphogenic defects in mice [24,25]. Cyclopamine was subsequently demonstrated to bind to and inhibit SMO [26], and based on these early findings, several additional SMO antagonists have been developed and tested clinically, including vismodegib and sonidegib, which are approved for advanced BCC, and glasdegib for AML.

Preclinical studies have demonstrated that SMO inhibitors, including cyclopamine, vismodegib, erismodegob/sonidegib, and glasdegib, are capable of inhibiting CML bulk cells and LSCs either alone or in combination with BCR-ABL tyrosine kinase inhibitors (TKIs) [11,12,20,21,27]. Based on these studies, two clinical trials have been reported in CML targeting the HH signaling pathway through SMO inhibition. The first study was a phase 1 trial examining the safety and efficacy of the SMO inhibitor BMS-833923 (XL-139; Bristol Myers Squib) in patients with chronic phase CML (CML-CP) who failed or responded sub-optimally to prior TKI or with accelerated phase CML (CML-AP) or Ph + acute lymphoblastic leukemia (ALL) resistant to imatinib or nilotinib [28]. Twenty-seven patients were enrolled and treated with the second-generation TKI dasatinib for 4 weeks followed by the addition of BMS-833923. The most frequent adverse events were like those seen with the first clinically tested SMO inhibitor vismodegib and consisted of dysgeusia, alopecia, anorexia of nausea, muscle spasms, and fatigue [29]. Given co-administration of dasatinib, the efficacy of BMS-833923 was difficult to definitively demonstrate, but the addition of the SMO inhibitor resulted in prolonged responses, including one complete cytogenetic response (CCyR) in three of ten CP-CML patients previously resistant to dasatinib.

A second trial examined the safety and efficacy of combining the TKI nilotinib and SMO inhibitor erismodegib/sonidegib (LDE225; Novartis) in a phase 1b clinical trial that included 11 CML-CP patients who were resistant, intolerant, or sub-optimally responding to one prior TKI, excluding nilotinib [30]. The most frequent toxicities observed were consistent with SMO inhibition, but notably two patients experienced grade 4 elevations in blood creatine phosphokinase (CPK) levels suggestive of muscle injury. Efficacy included the maintenance of CCyR at baseline for 12 months in eight of ten patients and major molecular responses (MMR) in three patients. Moreover, improvements in cytogenetic and molecular responses were seen in three additional patients. Overall, the activity of the combination was modest, and given the poor tolerability of the combination, this trial was stopped prior to the planned phase 2 expansion.

## 4. Hedgehog Signaling in AML

AML is characterized by the proliferation and accumulation of immature leukemic blasts within the bone marrow, which interferes with normal hematopoiesis. It is also molecularly heterogeneous with a diverse range of recurrent gene mutations and chromosomal alterations. Several studies have demonstrated that components of the HH signaling pathway are expressed in AML cell lines and clinical samples, including *PTCH*, *SMO*, and all three *GLI* transcription factors [31,32,33,34]. Furthermore, the expression of *GLI1* and *GLI2* suggestive of pathway activation in clinical specimens is associated with chemotherapeutic resistance and inferior survival rates, providing additional evidence that HH signaling is clinically relevant in patients with AML and myelodysplastic syndrome (MDS) [35,36,37,38]. Pharmacological and genetic inhibition of HH signaling in these cells has significant antileukemic effects, including the induction of apoptosis, reduced proliferation and colony formation in AML cells, and prolonged survival in mouse xenograft studies [21,31,39,40]. Furthermore, the inhibition of SMO can target LSCs and decrease tumor initiating potential [21,39].

AML is typically treated with high doses of cytotoxic chemotherapeutic agents, including anthracyclines (e.g., daunorubicin, idarubicin) and the nucleoside analogue cytarabine. Resistance to these drugs is a major challenge in the management of AML, and multiple studies have found that chemotherapeutic resistance is associated with HH pathway activation [31,34,41]. Active HH signaling promotes multi-drug resistance by inducing the expression of the P-glycoprotein transporters that export chemotherapeutic agents, and cyclopamine can re-sensitize AML cells to cytotoxic agents [42]. It can also induce drug resistance by modulating drug metabolism as GLI1 induces the expression of UDP glucuronosyltransferase (UGT1A), which inactivates drugs by glucuronidation in chemotherapy-resistant AML cells, and the inhibition of GLI1 activity either through siRNA-mediated knockdown or treatment with the SMO inhibitor vismodegib sensitizes AML cells to cytarabine or the eIF4E inhibitor ribavirin [34]. Similarly, the loss of GLI3 expression is associated with cytarabine resistance in AML cells and promotes the expression of several factors associated with drug transport and metabolism [43].

In patients unable to receive intensive chemotherapy, hypomethylating agents (HMAs), including azacytidine and decitabine, can be used alone or in combination with other targeted agents. In elderly AML patients, the combination of azacytidine and the BCL2 inhibitor venetoclax has become a standard approach based on superior response and survival rates compared to azaciticine alone [44]. The mechanisms responsible for HMA resistance are unknown, but an RNAi-based screen in AML cells identified several HH pathway components (*SMO*, *SHH*, and *GLI3*) that were associated with 5-azacytidine resistance, and the combination of the SMO inhibitor sonidegib with 5-Aza led to synergistic cell killing [40]. A second study demonstrated that *GLI1* silencing inhibited proliferation and induced apoptosis of MUTZ-1 cells derived from an MDS patient and enhanced the demethylation of the p15 promoter by 5-Aza [45]. Further studies into the interactions between HMA therapy and HH signaling have demonstrated that glasdegib induced the expression of GLI3R, which in turn inhibited the expression of AKT and AML cell viability [46]. These investigators also found that the GLI3R promoter is silenced by methylation in some cases of AML, but treatment with decitabine induced GLI3R expression and sensitized cells to SMO inhibition. Therefore, combining a HMA and SMO antagonist may represent a promising approach to AML, especially in those cases lacking GLI3R expression.

In addition to cytotoxic chemotherapeutic agents and HMAs that can be broadly used in AML, several drugs have been approved to target specific and recurrent mutations found in AML. FLT3 is a receptor tyrosine kinase that is mutated in approximately 30% of adult AML patients [47]. Point mutations within the kinase domain or internal tandem duplication (ITD) in the juxta-membrane domain result in constitutive kinase activity and are associated with poor outcomes. Multiple studies have demonstrated that a higher expression of *GLI2* is associated with inferior survival rates in FLT3-mutant AML [35,36,48], and in a novel mouse model, HH pathway activation through the expression of the constitutively active SMO mutant SMO-M2 resulted in the transformation of a myeloproliferative disorder generated by the FLT3-ITD mutation alone into AML [35]. The enhanced kinase activity of mutant FLT3 constitutively activated downstream Stat5, which directly induced *Gli2* expression. With the expression of *Gli2*, AML cells gain responsiveness to the HH ligand, resulting in pathway activation (Figure 1). In this and other studies, the combination of FLT3 inhibitors and SMO antagonists was found to inhibit AML growth in a synergistic manner, suggesting that combinations of kinase and HH pathway inhibitors may be clinically effective [48].

AML can both arise as de novo disease in healthy individuals and secondary acute leukemia in patients with chronic hematologic malignancies. The activation of HH signaling alone in the hematopoietic system does not lead to leukemia in mice and is unlikely to be an initiating event in myeloid malignancies [12,14]. In addition to the progression of the indolent myeloproliferative state in FLT3-ITD mice by constitutively active SMO [35], these findings suggest that aberrant HH signaling may primarily play a role in promoting secondary AML. A recent study examined a role of HH signaling during the progression of MDS and found that *GLI1* expression increased with the progression of MDS to secondary AML in matched clinical specimens [38]. In addition, conditional expression of SMO-M2 in the hematopoietic system of a mouse model of MDS driven by the NUP98-HOXD13 fusion gene led to progression to aggressive AML. Further analysis demonstrated that HH signaling induced the expansion of myeloid progenitors that expressed several self-renewal pathways and could propagate disease during serial transplantation indicative of LSC activity. These findings suggest that aberrant HH activity may primarily accelerate the progression and transformation of indolent myeloid diseases to AML.

## 5. Clinical Studies in AML

Multiple SMO inhibitors have been clinically studied in AML, either alone or in combination with additional anti-leukemic agents. Vismodegib (GDC-0449; Genentech, San Francisco, CA, USA) was modified from a benzimidazole compound identified in a high-throughput screen using a mouse embryonic fibroblast and a GLI reporter system and eventually became the first SMO inhibitor approved for clinical use [49,50]. A phase 1b trial evaluated the safety and efficacy of vismodegib in 38 relapsed or refractory AML patients [51]. Aside from the hematologic toxicities expected in AML, the most common adverse events were nausea and dysgeusia similar to the use of vismodegib in BCC patients. The efficacy of vismodegib was limited with an overall response rate (ORR) of 6.1%, which included one complete remission with incomplete hematologic recovery (CRi) and one partial response (PR). The median OS in all treated patients was 3.4 months.

Sonidegib was developed from a biphenyl carboxamide identified using a HH pathway reporter system in mouse testicular epithelial cells and has been FDA approved for the treatment of locally advanced BCC [52]. A phase 1/1b clinical trial examined the safety and efficacy of the combination of sonidegib and 5-Aza in 63 patients with newly diagnosed or relapsed/refractory myeloid malignancies, including AML, intermediate and high-risk MDS, chronic myelomonocytic leukemia (CMMoL), and myelofibrosis (MF) [53]. In terms of safety, non-hematologic adverse events were generally mild and most frequently consisted of fatigue, constipation, nausea, cough, and diarrhea. Overall response rates (ORR) for AML and MDS were 23.1% and 7.1% for newly diagnosed and relapsed/refractory patients, respectively. In addition, relapsed/refractory AML experienced a high rate of stable disease (SD) at 76% and a median overall survival of 7.6 months. A second trial examined the addition of sonidegib to 5-Aza in 23 high-risk MDS patients who failed to respond or lost response to 5-Aza alone [54]. During the initial dose escalation phase, grade 4 adverse events included elevations in CPK levels and aspartate aminotransferase (ASAT) levels consistent with hepatic toxicity, and within the complete cohort, infections occurred in 73% of all evaluable patients. The ORR was 13.6% and included one complete remission (CR), marrow CR, and PR, and the median overall survival was 6.8 months.

Glasdegib (PF-04449913; Pfizer, New York, NY, USA) is an oral SMO inhibitor [55], and seven clinical trials for myeloid malignancies have been reported, including four as a single agent and three in combination with other therapies. An initial dose-escalation phase I trial in myeloid malignancies examined single agent glasdegib in 47 patients with untreated or previously treated AML, MDS, CML, MF, and CMMoL, with the majority (60%) with AML [56]. Treatment-related non-hematologic adverse events were mostly grade 1 or 2 in severity, and the most common toxicities were consistent with SMO inhibition including dysgeusia, decreased appetite, and alopecia. Notably, grade 3 QTc prolongation was observed in four of five patients at the highest dose level of 600mg daily but resolved after treatment of reversable causes. Although the ORR was not described due to disease heterogeneity, potential responses were noted in 23 (49%) of the patients and included one CRi in a patient with secondary AML arising from CMMoL and four PRs (18%) in 28 patients with AML, hematologic improvement in two of six MDS patients, and PRs and SD in the remaining 16 responding patients with CML, MF, or AML. For the entire cohort of patients, the median progression free survival was 4.4 months.

A second clinical trial of single agent glasdegib consisted of a phase 2 study in patients with AML, MDS, and CMMoL previously failing HMA treatment [57]. The majority (74%) of the 35 patients enrolled and evaluable for treatment responses were diagnosed with MDS, and following a median of three (range 0–11) cycles, two (6%) patients experienced marrow CRs with improved peripheral blood counts. An additional 56% of patients had SD, and the median OS was 10.4 months. The spectrum and severity of non-hematologic toxicities were similar to those seen in the previous phase 1 study of glasdegib in myeloid malignancies. A separate phase 1 study of single agent glasdegib in Japanese patients with hematologic malignancies enrolled 13 patients with previously treated AML, MDS, CMMoL, and MF [58]. Toxicities were similar to other glasdegib studies, and efficacy consisted of one CR and four patients with SD in seven AML patients and one CR and two SD in four patients with MDS. Taken together, these studies suggest that glasdegib has moderate clinical activity as a single agent across a wide range of myeloid malignancies.

A fourth clinical trial using single agent glasdegib examined its use as maintenance therapy in 31 patients with AML or MDS following myeloablative or non-myeloablative allogeneic bone marrow transplantation (alloBMT) [59]. Glasdegib was initiated at a median of 46 days post-transplant and was administered daily for a median of 142 days, with 26% of patients completing the planned one year of maintenance therapy. The 1- and 2-year relapse-free survival (RFS) rates were 42% and 32%, respectively, and the cumulative incidence of relapse, including minimal residual disease (MRD) positivity at 1 year, was 45%. The 1- and 2-year OS rates were 65% and 47%, respectively. Grade 2–4 graft versus host disease (GVHD) and chronic GVHD occurred at a rate of 38.7 and 41.9%, respectively. Both the relapse and GVHD rates were similar to historical alloBMT data, suggesting that the effects of glasdegib were limited in this setting, but overall, frequent dose interruptions and reductions due to toxicity led to poor adherence.

Combination studies of glasdegib include a three-arm open label phase 1b dose-escalation trial in AML and high-risk MDS patients examining glasdegib with low-dose cytarabine (LDAC, Arm A) or the HMA decitabine (Arm B) in patients who were considered unsuitable for standard induction chemotherapy due to age, performance state, or renal or cardiac insufficiency. A third arm consisted of glasdegib in combination with daunorubicin and cytarabine in medically fit patients (Arm C) [60]. A total of 52 patients were enrolled, and the majority of patients (87%) were diagnosed with AML. An initial dose-escalation phase was carried out in each arm and demonstrated that these combinations, in general, were well tolerated and consistent with the toxicities of each anti-leukemic regimen. Toxicities associated specifically with SMO inhibitors were primarily grade 1 or 2 events, and a recommended phase 2 dose (RP2D) of 100mg daily was established based on these findings. Each arm was further expanded, and the overall CR + CRi rates in each cohort were 8.7% in Arm A, 28.6% in B, and 54% in C, with a median OS of 4.4, 11.5, and 34.7 months, respectively. In the AML patients, a clinically defined beneficial response that included PRs was observed in 10% of patients in Arm A, 60% in B, and 60% in C. For MDS, 67% of patients in Arm A and all patients in Arms B and C had either a CR or marrow CR.

A single-arm phase II trial of glasdegib and standard induction chemotherapy in newly diagnosed AML of MDS patients was subsequently carried out based on the RP2D based on the initial phase 1b combination study described above [61]. Glasdegib was administered during daunorubicin and cytarabine (7 + 3) induction chemotherapy, two to four cycles of consolidation with cytarabine, and maintenance therapy as a single agent for six months. Sixty-nine patients were enrolled on the study, and the majority (91%) did not complete the full course of therapy due to a lack of clinical response, referral to alloBMT, or treatment-related adverse events. Notably, 93% of the enrolled patients were diagnosed with AML and 87% of the evaluable patients were older than 55 years of age. The CR rate was 46% and the median OS was 14.9 months. The OS rates compared favorably with historical data from risk-adjusted cohorts treated with cytarabine-based intensive regimens. In general, the addition of glasdegib was well tolerated, with limited adverse events typically observed with SMO inhibitors.

The randomized, open-label, phase 2 BRIGHT AML 1003 trial in newly diagnosed AML and high-risk MDS patients with co-morbidities that prevented the use of intensive induction chemotherapy was also developed following the initial phase 1b combination study [62]. In this trial, 122 patients were randomized to receive low-dose cytarabine (LDAC) with or without glasdegib in a 2:1 ratio. In 115 evaluable patients, the combination of glasdegib and LDAC (G-LDAC) produced a significant difference in CR rate of 17% compared to 2.3% for LDAC alone. Furthermore, the primary endpoint of the trial, median OS, significantly favored G-LDAC at 8.8 months compared to 4.9 months with LDAC alone. The majority (94%) of patients enrolled in the trial were diagnosed with AML, and among these, the median OS was also significantly different for the G-LDAC and LDAC arms at 8.3 months and 4.3 months, respectively. Based on this OS benefit, the G-LDAC combination was approved by the FDA for clinical use in newly diagnosed AML patients unfit for intensive induction chemotherapy in 2018. With additional follow up, the OS advantage for G-LDAC was not limited to patients who achieved a CR as the median OS of G-LDAC-treated patients without CR or CRi was 5.0 vs. 4.1 months [63]. In the HR-MDS patients, G-LDAC was associated with a 22.8% reduction in the risk of death compared to LDAC alone, but the small number of MDS patients limited the approval of G-LDAC to AML. An additional two phase 3 studies are currently ongoing in the BRIGHT AML 1019 study that is examining glasdegib or placebo in combination with either cytarabine and daunorubicin or azacytidine in patients with untreated AML [64].

## 6. GLI Activation and Targeting in Myeloid Malignancies

In several solid tumors, the GLI transcription factors can be activated downstream of SMO by several commonly activated oncogenic pathways such as KRAS, PI3 kinase, mTOR, and TGF-β [65]. Cross talk with these other pathways has been found to induce and enhance GLI1 and in some cases GLI2 activity through multiple mechanisms including gene expression, cellular localization, protein stability, and transcriptional activity. Since strategies targeting the HH pathway through SMO inhibition alone are unlikely to be effective in these cases, additional targeting strategies are needed. In myeloid malignancies, the MDS cell MDS-L has been found to be resistant to glasdegib, but the inhibition of GLI1 by siRNA-mediated knockdown or the GLI1 antagonist, GANT61, inhibits cell proliferation [38]. Therefore, approaches that directly target GLI may be useful in myeloid malignancies, and in Figure 2, we provide a schema of potential strategies.

Several studies have demonstrated that GLI1 expression can be inhibited by targeting the oncogenic pathways that lead to its activation. For example, aberrant MEK1 activity within the MAPK signaling pathway leads to *GLI1* expression, which can be subsequently inhibited by pharmacologic MEK inhibition [66]. One potential mechanism driving GLI1 activity involves Suppressor of Fused (SUFU), a negative regulator of HH signaling that retains GLI1 in the cytoplasm, and increased MAPK signaling has been found to inhibit SUFU and allow the nuclear translocation of GLI1 [67]. Similarly, active PI3K signaling leads to GLI1 phosphorylation at serine 84 by S6 Kinase 1 (S6K1), which dissociates GLI1 from SUFU [68]. TGF-β signaling can activate HH signaling by directly inducing *GLI2* expression through SMAD3 and β-catenin binding to the *GLI2* promoter [69]. Multiple inhibitors have been developed to target each of these pathways, and these findings may provide the basis for their use as GLI1 inhibitors.

GLI1 is primarily regulated by its gene expression, but it also undergoes multiple post-translational modifications and protein interactions that may serve as novel targeting approaches. In addition to S6K1, other kinases can phosphorylate GLI1 and impact its activity. Atypical Protein Kinase C (aPKC, PKC iota/lambda) levels can be increased in SMO inhibitor-resistant BCC cells and can phosphorylate GLI1 near its zinc-finger DNA binding domain to increase its transcriptional activity [70]. The family of Dual-Specificity Tyrosine Phosphorylation Regulated Kinases (DYRK) can also phosphorylate and activate GLI1 by promoting nuclear translocation [71,72]. This effect appears to be primarily mediated by DYRK1B, and DYRK1B has been found to inhibit the growth of GLI-dependent pancreatic cancer cells [73]. On the other hand, additional DYRK family members, specifically DYRK2, have been found to inhibit HH signaling by phosphorylating GLI2, which leads to its degradation [74]. Therefore, targeting DYRK may require inhibiting DYRK1B but not DYRK2. Similarly, the activation of Casein kinase 1-alpha (CK1A) leads to the preferential formation of GLI repressor isoforms and GLI degradation, but at the same time also promotes GLI accumulation within primary cilia that may enhance GLI activity [75].

The deacetylation of GLI1 and GLI2 is required for their full activity [76], and histone deacetylase (HDAC) inhibitors can decrease their transcriptional activity [77]. HDAC inhibitors are approved for the treatment of T cell lymphomas and multiple myeloma, suggesting that these agents could inhibit GLI activity in the clinical setting either alone or in combination with other GLI targeting agents, such as aPKC inhibitors [78,79]. The transcriptional activity of GLIs can be modulated by the BET family member Bromodomain 4 (BRD4), which binds directly to the GLI1- and GLI2-dependent promoters [80,81], and several BRD4 inhibitors have entered clinical testing and may be novel strategies to target SMO-independent GLI activation. Several BET inhibitors are undergoing early clinical evaluation and may serve as GLI inhibitors.

Additional agents that can inhibit aberrant GLI activation include arsenic trioxide (ATO), which is approved for the treatment of acute promyelocytic leukemia. ATO has been found to inhibit HH signaling by preventing ciliary accumulation of GLI2 and the growth of tumors with acquired resistance to SMO antagonists [82]. ATO can also interact with GLI1 to inhibit its transcriptional activation of HH target genes [83], suggesting that it may target the HH signaling pathway at multiple points downstream of SMO. Oxysterols are oxidized derivatives of cholesterol that have multiple effects on the HH signaling pathway. These compounds can act as both SMO agonists and antagonists as well as act as ligands for Liver X Receptors (LXR), which can inhibit GLI1 activity downstream of SMO [84,85,86]. Taken together, several strategies may be capable of targeting SMO-independent GLI activation.

## 7. Summary and Future Directions

The HH signaling pathway is widely active in myeloid malignancies and promotes tumor cell proliferation, drug resistance, and LSC properties. Early clinical studies using SMO antagonists have demonstrated promising data, and in the case of glasdegib, have been approved for use in AML based on improved OS rates. However, the activity of SMO inhibitors has been limited in most other trials, and coupled with the toxicities common to all of these compounds, alternatives approaches are needed. Since ligand- and SMO-independent GLI activation has been documented in AML and MDS, novel approaches capable of directly targeting GLI may expand the utility of HH pathway inhibition in these diseases as well as other cancers.

## Figures and Tables

**Figure 1 cancers-13-04888-f001:**
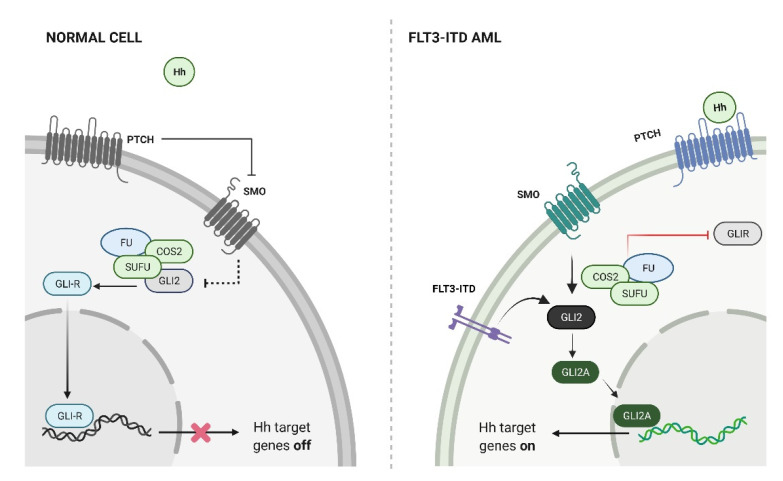
HH pathway activation in FLT3-ITD AML. Constitutive FLT3-ITD kinase activity induces the expression of GLI2, which permits responsiveness to HH ligands (illustrated using BioRender.com, accessed on 14 June 2021).

**Figure 2 cancers-13-04888-f002:**
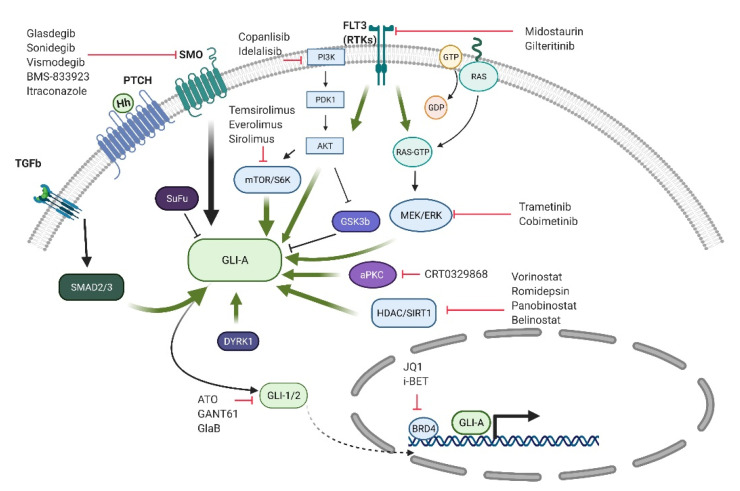
SMO-independent GLI activation. Several oncogenic pathways are capable of activating GLI downstream of SMO. Potential therapeutic strategies and their targets are listed (illustrated using BioRender.com, accessed on 14 June 2021).

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
