# Peer review of "Hedgehog Signaling in Myeloid Malignancies"

_cancers, 2021, doi:10.3390/cancers13194888_

Round 1
Reviewer 1 Report
The article is well written and I find especially relevant the summary of clinical studies of SMO inhibitors in AML.
- The phrase on lines 59-61 is confusing. Please check and re-word.
- The phrases on lines 87-95 and 113-123 are included in the CML subsection, but they don't fit here.
- Glasdegib was approved by the FDA in 2018. Could the authors briefly discuss some real-world results on its use.
Author Response
In response to Reviewer #1, we have rewritten this sentence from line 59-61 and deleted the two sentences in the CML section. For Reviewer #2, we have rewritten this sentence in response to Reviewer #1, added a sentence regarding SMO inhibition and hematologic toxicity in section 2. We are not clear what specifically the Reviewer is referring to in section 4 and have removed the sentence previously at lines 224-226. We have made the change requested by Reviewer #3.
Reviewer 2 Report
#Section 2
1.- Authors explain that “the precise role of Hh signalling in adult haematopoiesis appears to be highly dependent on the specific context in which it is examined”. Maybe, it should be highlighted this discrepancy. Therefore, could the authors please discuss the possible repercussions of the inhibition of the Hh pathway in non-malignant cells. For example, what would happen if Hh signalling is needed for adult haematopoiesis.
#Section 3
2.- The line break between 123-124 and 140-141 seems incorrect.
#Section 4
3.- The role of Hh inhibition in LSCs is contradictory. Although Fukushima and Sadarangani postulates that the inhibition of Hh reduce the quiescence of LSC, their works only show the CD45 population. This population correspond also with progenitors. Therefore, more studies must be performed in order to isolate the LSC when Hh is inhibited. Other reviews related the complexity and the controversy of LSCs, such as:
Carter, J.L., Hege, K., Yang, J. et al. Targeting multiple signaling pathways: the new approach to acute myeloid leukemia therapy. Sig Transduct Target Ther 2020, 5, 288 (2020). https://doi.org/10.1038/s41392-020-00361-x
Lainez-González, D.; Serrano-López, J.; Alonso-Domínguez, J.M. Understanding the Hedgehog Signaling Pathway in Acute Myeloid Leukemia Stem Cells: A Necessary Step toward a Cure. Biology 2021, 10, 255. https://doi.org/10.3390/biology10040255
4.- Does the lines 224-226 suit better in the section focus on CML instead AML?
Author Response

(The authors gave the same response as above.)

Reviewer 3 Report
This manuscript is a review covering from the minimal basic framework of the Hedgehog signaling in hematopoiesis to the clinical studies with its inhibitors for myeloid leukemias. I think that it is well-written with the adequate references of recent important studies.
One part, however, is apparently wrong and should be corrected (Page 9, line 404). “The acetylation of GLI1 and GLI2 is ….” is not correct and “acetylation” should be replaced by “deacetylation”. Otherwise, one cannot understand why HDAC inhibitors can decrease their transcriptional activity.
Author Response

(The authors gave the same response as above.)

Reviewer 4 Report
The authors provide a comprehensive review on the hedgehog pathway including its role in embryogenesis, normal and malignant hematopoiesis. Furthermore, they discuss current clinical trials in AML and CML employing smoothened inhibitors. In general, the review is well written and up-to-date.
But, in the basic science part, I would like to have two additional points discussed.
1) Canonical Hedgehoog signalling requires the presence of primary cilia, at least in epithelial cells. But in AMLs cell, primary cilia are rare and dysmorphic (Singh M et al., Exp Hematol 2016). This observation may explain why the hedgehog pathway is frequently activated non-canonically. The reduced frequency and dysfunction of ciliae may result in reduced sensitivity to smoothened inhibitors.
2) In hematological neoplasms hedgehog signalling may not be restricted to leukemic cells, but may also occur in bone marrow niche cells (Kramann R et al. Blood 2016, Klein C et al. J exp Med 2016). This may contribute the pathology of these diseases. Inhibition of hedgehog signalling in niche cells may be an important part of the action of smoothened inhibitors.
Author Response

(The authors gave the same response as above.)

Round 2
Reviewer 2 Report
The review is well focused, overall, in Hedgehog treatments in AML.
In this last version some improvements have been done. Nevertheless, authors claim that they have removed the sentence at lines 224-226 (v.1) but it is now in lines 213-215 (v.2): "In CML, the expression of HH target genes, including GLI1 and PTCH1, is low in leukemia cells isolated from patients in CML-CP, but increases during progression to CML-AP and CML-BC". I still believe this sentence fits better in the CML section.
Moreover, as previously explained, the inhibition of Hh in LSC are contradictory. The studies of Fukushima and Sadarangani only focused in the CD45 population, while LSC are described, mostly, as CD45dim/CD34+/CD38- pool. The sentence “the inhibition of SMO can target LSCs and decrease tumor initiating potential” may asseverate that they studied the LSCs instead the CD45 population. Therefore, different conclusion can be withdrawn. It could be more accurate to those studies some though as: “Although the authors explain that their results focused in LSC population, they only study the CD45 pool. Nonetheless, they proved that in this population the inhibition of SMO decreases the tumour initiating potential”.
Author Response
- The sentence "In CML, the expression of HH target genes, including GLI1 and PTCH1, is low in leukemia cells isolated from patients in CML-CP, but increases during progression to CML-AP and CML-BC" has now been moved from its previous position at line 213 to the CML section at line 126.
2. We have further clarifies the sentence that includes "the inhibition of SMO can target LSCs and decrease tumor initiating potential” to "The role of SMO and HH signaling directly in LSCs is not completely clear as studies have demonstrated that SMO inhibtion can decrease tumor initiating potential but direct examination of LSCs has relied on surface antigens that may or may not be optimal or consistent from patient to patient [21,39]."
Round 3
Reviewer 2 Report
Authors made considerable improves in the manuscript. Although the role of LSCs is not clear and some of the works they citated study the population CD45 instead CD34/CD38.